# Convex Elicitation of Continuous Real Properties

**Jessica Finocchiaro**
Department of Computer Science
University of Colorado, Boulder
jessica.finocchiaro@colorado.edu

**Rafael Frongillo**
Department of Computer Science
University of Colorado, Boulder
raf@colorado.edu

## Abstract

A property or statistic of a distribution is said to be elicitable if it can be expressed as the minimizer of some loss function in expectation. Recent work shows that continuous real-valued properties are elicitable if and only if they are identifiable, meaning the set of distributions with the same property value can be described by linear constraints. From a practical standpoint, one may ask for which such properties do there exist convex loss functions. In this paper, in a finite-outcome setting, we show that in fact essentially *every* elicitable real-valued property can be elicited by a convex loss function. Our proof is constructive, and leads to convex loss functions for new properties.

## 1   Introduction

Property elicitation is the study of statistics, or *properties*, of probability distributions which one can incentivize an expected-utility-maximizing agent to reveal. In a machine learning context, this "agent" is an algorithm following the principle of empirical risk minimization (ERM), wherein a hypothesis is fit to the data by minimizing its error on a training data set, as judged by some loss function. The interest in property elicitation across the economics, statistics, and machine learning communities is reflected in the literature, with important results appearing in all three.

A central thread of this literature, weaving between all three communities, asks which continuous real-valued properties are elicitable, and which loss functions elicit them. Building on earlier work of Osband [16] and Lambert [12], Steinwart et al. [22] show that a property is elicitable if and only if it is *identifiable*, a concept introduced by Osband which says that the set of distributions sharing the same property value can be described by a set of linear constraints. Moreover, these papers give characterizations of the loss functions eliciting these identifiable properties, showing that every loss can be written as the integral of a positive-weighted identification function.

A question of practical interest remains, however: for which properties do there exist *convex* loss functions eliciting them? Convex losses give concrete algorithms to efficiently solve ERM problems, and are also useful more broadly in statistical and economic settings (see § 6). At first glance, the answer to this question might appear to follow immediately from the comprehensive loss function characterizations of Lambert [12] and Steinwart et al. [22]. Unfortunately, it is far from clear in these characterizations whether there exists a weight function rendering their construction convex.

In this paper, we address this question of convex elicitability in the finite-outcome setting. Surprisingly, we find that, under somewhat mild smoothness assumptions, *every* identifiable real-valued property is convex elicitable. Our proof proceeds by pinpointing a few key attributes of identification functions, and then solving the following abstract problem: given a set of functions $\mathcal{F}$ from $\mathcal{R}$ to $\mathbb{R}$, when does there exist a weight function $\lambda : \mathcal{R} \to \mathbb{R}_{>0}$ making $\lambda f$ increasing over the report space $\mathcal{R}$ for all $f \in \mathcal{F}$? We give a constructive solution to this problem under certain conditions, and show that identification functions happen to satisfy these conditions.

After reviewing the relevant prior work in more detail (§ 2), we give our main result (§ 3). We then give intuition for our key technical proposition, the solution to the abstract problem mentioned above (§ 4), followed by examples illustrating the constructive nature of our approach (§ 5). We conclude with a discussion of applications to information elicitation, and future work (§ 6). See the Appendix for all omitted proofs.

**Notation.** We will use the following notation throughout the paper. Let $\mathbb{R}_{>0} := \{r : r \in \mathbb{R}, r > 0\}$ denote the positive reals. For an interval $\mathcal{I}$, let $\mathring{\mathcal{I}}$ denote the interior of $\mathcal{I}$. We let $\mathcal{Y}$ denote the outcomes space, here taken to be finite, and $\Delta(\mathcal{Y})$ denote the set of probability distributions over $\mathcal{Y}$.

## 2    Setting and Background

In property elicitation, we aim to learn some distributional property by minimizing a loss function. A *property* is simply a function $\Gamma : \mathcal{P} \to \mathcal{R}$, which assigns a desired report $\mathcal{R} \subseteq \mathbb{R}^k$ to each probability distribution in a convex set $\mathcal{P} \subseteq \Delta(\mathcal{Y})$. Without loss of generality, we often restrict $\mathcal{R} = \Gamma(\mathcal{P})$, but this does not affect the result. Common properties include moments, quantiles, and expectiles. Throughout the paper we will assume that $\Gamma$ is a continuous real-valued function, implying $\mathcal{R} \subseteq \mathbb{R}$ is an interval. We also restrict to the finite outcome setting, where $|\mathcal{Y}| < \infty$, and consider $\mathcal{P} \subset \mathbb{R}^{|\mathcal{Y}|}$, meaning we identify each distribution with the corresponding vector of probabilities.

We are interested in when properties are *elicitable*, meaning they can be expressed as the minimizer of expected loss for some loss function. In the present paper, we will additionally ask when the loss function can be convex.

**Definition 1.** *A loss function* $L : \mathcal{R} \times \mathcal{Y} \to \mathbb{R} \cup \{\infty\}$ *elicits a property* $\Gamma$ *if for all* $p \in \mathcal{P}$,

$$\{\Gamma(p)\} = \arg\min_r \mathbb{E}_{Y \sim p} L(r, Y) . \tag{1}$$

*In this case, we say* $\Gamma$ *is* elicitable. *If* $L(\cdot, y)$ *is convex for every* $y \in \mathcal{Y}$, *we say* $\Gamma$ *is* convex elicitable.

A central notion in property elicitation is that of identifiability, where the level sets $\{p : \Gamma(p) = r\}$ can be expressed as a linear constraint.

**Definition 2.** *Let property* $\Gamma : \mathcal{P} \to \mathcal{R}$ *be given, where* $\mathcal{R} = \Gamma(\mathcal{P})$. *A function* $V : \mathcal{R} \times \mathcal{Y} \to \mathbb{R}$ *identifies* $\Gamma$ *if*

$$\mathbb{E}_{Y \sim p}[V(r, Y)] = 0 \iff r = \Gamma(p) \tag{2}$$

*for all* $r \in \mathring{\mathcal{R}}$ *and* $p \in \mathcal{P}$. *In this case we say* $\Gamma$ *is* identifiable. *We say* $V$ *is* oriented *if we additionally have* $\mathbb{E}_{Y \sim p}[V(r, Y)] > 0 \iff r > \Gamma(p)$, *for all* $r \in \mathring{\mathcal{R}}$ *and* $p \in \mathcal{P}$.

Note that by the terminology of Steinwart et al. [22], an identification satisfying eq. (2) on all of $\mathring{\mathcal{R}}$ is called *strong*, as otherwise it must hold almost everywhere. We can loosely think of an identification function as a derivative of a loss; if $L$ is differentiable and elicits $\Gamma$, then roughly speaking, we expect $\frac{d}{dr}\mathbb{E}_{Y \sim p}L(r, Y) = 0 \iff \Gamma(p) = r$.

Finally, we will often assume our properties to possess two important qualities: continuity, and being nowhere-locally-constant.

**Definition 3** (Lambert [12]). *A continuous property* $\Gamma : \mathcal{P} \to \mathcal{R}$ *is* nowhere-locally-constant *if there does not exist any open neighborbood* $U$ *of* $\mathcal{P}$ *such that* $\Gamma(p) = r$ *for all* $p \in U$.

Intuitively, restricting to nowhere-locally-constant properties is merely to ease bookkeeping, as one could always collapse different report values together afterwards.

It is known that for continuous, nowhere-locally-constant, real-valued properties, identifiability implies elicitability. In this paper, we show that under slightly stronger assumptions, identifiability implies *convex* elicitability. To place this result in the proper context, we now briefly tour the history of property elicitation.

### 2.1    Relevant prior work

While Savage [20] studied the elicitation of expected values, the literature on the elicitation of general properties began with Osband [16], who gave several important results. One of Osband's observations is that the level sets $\{p : \Gamma(p) = r\}$ of an elicitable property $\Gamma$ must be convex [16,

Proposition 2.5]. He also introduced the notion of an identification function, and the so-called *Osband's principle*, which states that (under a mild regularity assumption) every loss function eliciting a given property can be written as the integral of a weighted identification function [16, Theorem 2.1]. He also gave several other results, such as the separability of loss functions jointly eliciting quantiles.

Independent of Osband, Lambert et al. [14, 15, 12] provide a geometric approach to both continuous and finite properties (that is, properties taking values in a finite set $\mathcal{R}$) when the set of outcomes $\mathcal{Y}$ is finite. They represent the identification function as a vector, and relating finite-valued properties to *power diagrams* in computational geometry. They rediscover several results of Osband for the real-valued case, such as convexity of level sets and a one-dimensional version of Osband's principle. Moreover, the proof of [12, Theorem 5] shows the following. (Steinwart, et al. [22] extend this result to the case of infinite $\mathcal{Y}$.)

**Theorem 1** (Lambert [12]). *Let $\Gamma : \mathcal{P} \to \mathbb{R}$ be a continuous, nowhere-locally-constant property. If the level sets $\{p \in \mathcal{P} : \Gamma(p) = r\}$ are convex, then $\Gamma$ is elicitable, and has a continuous, bounded, and oriented identification function.*

None of the above-mentioned papers address the question of when the loss eliciting the property in question is convex. This question has arisen in the context of surrogate risk minimization, where unlike our setting, one seeks to *indirectly* elicit a given finite-valued property (such as the mode, or most likely label) by first eliciting a real-valued property, and then applying a link function [5, 23]. For example, support vector machines (SVMs) commonly use hinge loss and then apply the sign function, a combination which indirectly elicits the mode [21]. This literature is related to *elicitation complexity*, where one asks how many dimensions of the surrogate property are needed so that the desired property can be computed via the link function [7, 9, 14]. This relationship was perhaps first identified by Agarwal and Agarwal [3], who restate prior work in terms of property elicitation, and specifically focus on convex losses. Finally, Reid, Vernet, and Williamson [18, 24] consider losses which indirectly elicit the full distribution, and consider convexity of the composite loss.

In contrast to this line of work, we seek the *direct* elicitation of *continuous* properties. While convex losses are well-known for several continuous properties of interest, including the mean and other expected values (squared loss), ratios of expectations (weighted squared loss), and the median and other quantiles (pinball loss), to our knowledge, to date there have been no results on the direct convex elicitation of general continuous properties.

## 3 Main Result

We will show that, under mild conditions, every elicitable real-valued property is also convex elicitable. Let us first give some intuition why one might suspect this statement to be true. From a geometric perspective, the level sets $\{p : \Gamma(p) = r\}$ of continuous elicitable properties are hyperplanes intersected with $\mathcal{P}$. As one changes $r$, the level sets may be locally parallel, in which case the property is locally a link of a linear property (expected value), or the level sets may not be parallel, in which case the property locally resembles a link of a ratio of expectations. In fact, the second case also covers the first, so we can say that, roughly speaking, every continuous property looks locally like a ratio of expectations. The following proposition states that if the property can actually be written as a finite piecewise ratio of expectations, it is convex elicitable. Hence, taking the limit as one approximates a given property better and better by ratios of expectations, one may suspect that indeed every continuous property is convex elicitable.

**Proposition 1.** *Continuous piecewise ratio-of-expectation properties are convex elicitable.*

*Proof.* First, we formalize the statement. Recall that $\mathcal{Y}$ is a finite set. Let $\phi_i : \mathcal{Y} \to \mathbb{R}$ and $\psi_i : \mathcal{Y} \to \mathbb{R}_{>0}$ be arbitrary for $i = 1, \ldots, k$, and let $\gamma_i(p) = \mathbb{E}_{Y \sim p} \phi_i(Y) / \mathbb{E}_{Y \sim p} \psi_i(Y)$. Assume that we have $a_0 < \cdots < a_k$ such that for all $p \in \mathcal{P}$, there is a unique $i \in \{1, \ldots, k\}$ such that either $\gamma_{i-1}(p) \in (a_{i-1}, a_i)$ or $\gamma_{i-1}(p) = \gamma_i(p) = a_{i-1}$. Call this $i(p)$, and by extension $i(r)$ where $r = \gamma_i(p)$ for this $i$. We will show that $\Gamma(p) := \gamma_{i(p)}(p)$ is convex elicitable with respect to the full probability simplex $\mathcal{P} = \Delta(\mathcal{Y})$.

Observe that by construction, for each $i \in \{1, \ldots, k - 1\}$ the level sets for $a_i$ coincide: $S_i = \{p : \Gamma(p) = a_i\} = \{p : \gamma_i(p) = a_i\} = \{p : \gamma_{i-1}(p) = a_i\}$. Moreover, for all such $i$, these level sets

are full-dimensional in $\mathcal{P}$, i.e., there are $(n-2)$-dimensional affine sets which are the intersection of a hyperplane and $\mathcal{P}$. Now let $V_i(r,y) = \psi_i(y)r - \phi_i(y)$, which identifies $\gamma_i$, and is strictly increasing in $r$ as $\psi_i(y) > 0$ for all $y$. We now see that the hyperplane which is the span of $S_i$ in $\mathbb{R}^n$ is orthogonal to the vectors $V_{i-1}(a_i, \cdot) \in \mathbb{R}^n$ and $V_i(a_i, \cdot) \in \mathbb{R}^n$, by the definition of identifiability. We conclude that there is some coefficient $\alpha_{i-1}$ such that $V_{i-1}(a_i, y) = \alpha_{i-1}V_i(a_i, y)$ for all $y \in \mathcal{Y}$. (In fact, $\alpha_{i-1} > 0$, as the coefficient of $r$ must be positive.) We then construct $\beta_{i(r)} = \prod_{j=0}^{i(r)} \alpha_j$ and write the identification as $V(r, y) = \beta_{i(r)}V_{i(r)}(r, y)$. $\qquad\square$

Moving now to the formal result, let $\mathcal{I} \subseteq \mathbb{R}$ be an interval. Our main technical ingredient shows, given a collection $\mathcal{F}$ of functions $f : \mathcal{I} \to \mathbb{R}$ satisfying certain conditions, how to construct a multiplier $\lambda : \mathcal{I} \to \mathbb{R}_{>0}$ making $\lambda f$ strictly increasing on $\mathring{\mathcal{I}}$ for all $f \in \mathcal{F}$. In our proof, the family $\mathcal{F}$ will be the set of identification functions $\{V(\cdot, y)\}_{y \in \mathcal{Y}}$, and $\lambda$ will play the role of the weight function in previous work ([16, Theorem 2.1], [11, Theorem 2.7], [14, Theorem 3]) showing that $L(r, y) = \int_{r_0}^{r} \lambda(x)V(x, y)dx$ elicits $\Gamma$. As $\lambda f$ is increasing, $L$ will additionally be convex.

Therefore, the conditions below are only mildly stronger than what Lambert shows to be true of the desired properties. We begin with our three conditions; the first we will assume, and the second and third we will prove hold for any oriented identification function.

**Condition 1.** *Every $f \in \mathcal{F}$ is continuous on $\mathring{\mathcal{I}}$, and continuously differentiable on $\mathring{\mathcal{I}}$ except on a finite set $S_f \subsetneq \mathring{\mathcal{I}}$. When $f$ is differentiable, $\frac{d}{dx}f(x)$ is finite. Additionally, if $x \in \mathring{\mathcal{I}}$ and $f(x) = 0$, then for all $z$ in some open neighborhood $U$ of $x$, $\frac{d}{dz}f(z) \geq 0$ whenever $f$ is differentiable.*

**Condition 2.** *Every $f \in \mathcal{F}$ is bounded and has at most one zero $x_f \in \mathring{\mathcal{I}}$ so that if $x_f$ exists, $f(x) < 0$ for $x < x_f$ and $f(x) > 0$ for $x > x_f$. If $f$ does not have a zero on $\mathring{\mathcal{I}}$, then either $f(x) < 0$ or $f(x) > 0$ for all $x \in \mathring{\mathcal{I}}$. For all $x \in \mathring{\mathcal{I}}$, at least one function $f \in \mathcal{F}$ is nonzero at $x$.*

**Condition 3.** *For all $f, g \in \mathcal{F}$ and all open subintervals $\mathcal{I}' \subseteq \mathring{\mathcal{I}}$ such that $f > 0 > g$ on $\mathcal{I}'$, the function $\frac{g}{f}$ is strictly increasing on $\mathcal{I}'$.*

Our main technical tool follows; we sketch the proof in § 4 and defer the full proof to the Appendix.

**Proposition 2.** *If $\mathcal{F}$ satisfies Condition 1, 2, and 3, then there exists a function $\lambda : \mathcal{I} \to \mathbb{R}_{>0}$ so that $\lambda f$ is increasing over $\mathring{\mathcal{I}}$ for every $f \in \mathcal{F}$.*

With this tool in hand, we are ready to prove our main result.

**Theorem 2.** *For $\mathcal{P} = \Delta(\mathcal{Y})$, let $\Gamma : \mathcal{P} \to \mathcal{R}$ be a continuous, nowhere-locally-constant property which is identified by a bounded and oriented $V : \mathcal{R} \times \mathcal{Y} \to \mathbb{R}$. If $\mathcal{F} = \{V(\cdot, y)\}_{y \in \mathcal{Y}}$ satisfies Condition 1, then $\Gamma$ is convex elicitable.*

*Proof.* We have assumed all $f \in \mathcal{F} = \{V(\cdot, y)\}_{y \in \mathcal{Y}}$ are bounded, oriented, and satisfy Condition 1, and thus to apply Proposition 2, we need only establish Conditions 2 and 3. A fact we use throughout is that $V(r, y) = \mathbb{E}_{Y \sim \delta_y} V(r, Y)$, where $\delta_y$ is the point distribution on $y \in \mathcal{Y}$.

To establish Condition 2, we procede in order. First, boundedness of each $f \in \mathcal{F}$ follows by assumption. Second, we show that each $f$ has at most one zero on $\mathring{\mathcal{R}}$. As $V$ identifies $\Gamma$, note that $V(r, y) = 0 \iff \Gamma(\delta_y) = r$ when $r \in \mathring{\mathcal{R}}$. As $\Gamma$ is single-valued, there can be at most one such $r \in \mathring{\mathcal{R}}$. Third, we must show that if $f$ has a zero on $\mathring{\mathcal{R}}$, it changes sign from negative to positive at that zero, and if not, $f$ never changes sign on $\mathring{\mathcal{R}}$. The first case follows from the fact that $\Gamma(\delta_y) = r$ and that $V$ is oriented. For the second case, $V(\cdot, y)$ has no zero on $\mathring{\mathcal{R}}$, and thus by continuity of $V$, cannot change sign on $\mathring{\mathcal{R}}$. Fourth, to see that $\mathcal{F}$ has at least one nonzero function for all $r \in \mathring{\mathcal{R}}$, note that if $V(r, y) = 0$ for all $y \in \mathcal{Y}$, then $\mathbb{E}_{Y \sim p} V(r, Y) = 0$ for all $p \in \mathcal{P}$. Thus, as $V$ identifies $\Gamma$ and $r \in \mathring{\mathcal{R}}$, we would have $\Gamma(p) = r$ for all $p$, contradicting nowhere-locally-constancy.

For Condition 3, consider $V(\cdot, y_0), V(\cdot, y_1) \in \mathcal{F}$ and open interval $\mathcal{I}' = (a, b)$ such that $V(r, y_0) > 0 > V(r, y_1)$ for all $r \in \mathcal{I}'$. We define $p_\alpha = (1 - \alpha)\delta_{y_0} + \alpha\delta_{y_1}$ and $\gamma(\alpha) = \Gamma(p_\alpha)$ for $\alpha \in [0, 1]$. Since $\Gamma$ is continuous and nowhere-locally-constant, [22, Cor. 9] implies that $\Gamma$ is quasi-monotone, which in turn implies that $\gamma$ is nondecreasing on $[0, 1]$.

We first show $\mathcal{I}' \subseteq \gamma([0,1]) = [\gamma(0), \gamma(1)]$. By definition of $\mathcal{I}'$, we know $r \in \mathcal{I}' \implies V(r, y_1) < 0 < V(r, y_0)$ and the orientation of $V$ then implies $\Gamma(\delta_{y_1}) > r > \Gamma(\delta_{y_0})$. Thus, $\Gamma(\delta_{y_1}) = \gamma(1) \geq b > a \geq \Gamma(\delta_{y_0}) = \gamma(0)$, with the strict inequality since $\mathcal{I}'$ is nonempty. We then see that $r \in (a, b) \implies r \in [\Gamma(\delta_{y_0}), \Gamma(\delta_{y_1})] = \gamma([0,1])$, and therefore $\mathcal{I}' \subseteq \gamma([0,1])$

Next, we show that $\gamma$ is not only nondecreasing but strictly increasing on $A = \gamma^{-1}(\mathcal{I}')$. Note that $A$ is itself an open interval as $\gamma$ is continuous. Let $\alpha, \alpha' \in A$, and suppose for a contradiction that $\gamma(\alpha) = \gamma(\alpha') = r \in \mathcal{I}' \subseteq \mathring{\mathcal{R}}$. Then $\Gamma(p_\alpha) = \Gamma(p_{\alpha'}) = r$, and as $V$ identifies $\Gamma$, we have $\mathbb{E}_{Y \sim p_\alpha} V(r, Y) = \mathbb{E}_{Y \sim p_{\alpha'}} V(r, Y) = 0$. Thus, $\mathbb{E}_{Y \sim p_0} V(r, Y) = (\alpha' \mathbb{E}_{Y \sim p_\alpha} V(r, Y) - \alpha \mathbb{E}_{Y \sim p_{\alpha'}} V(r, Y))/(\alpha' - \alpha) = 0$, and similarly for $p_1$. By identifiability again, we must now have $\Gamma(p_0) = \Gamma(p_1) = r$, contradicting $\Gamma(p_0) < \Gamma(p_1)$ as observed above.

Since $V$ identifies $\Gamma$, we have for $\alpha \in A$,

$$0 = \mathbb{E}_{Y \sim p_\alpha} V(\gamma(\alpha), Y) = (1 - \alpha) \mathbb{E}_{Y \sim \delta_{y_0}} \left[ V(\gamma(\alpha), Y) \right] + \alpha \mathbb{E}_{Y \sim \delta_{y_1}} \left[ V(\gamma(\alpha), Y) \right]$$
$$= (1 - \alpha) V(\gamma(\alpha), y_0) + \alpha V(\gamma(\alpha), y_1) \,,$$

from which we conclude the function $F(\alpha) = V(\gamma(\alpha), y_1)/V(\gamma(\alpha), y_0) = (\alpha - 1)/\alpha = 1 - 1/\alpha$, which is strictly increasing in $\alpha$. Observe that as $\gamma$ is strictly increasing on $A$, its inverse is strictly increasing on $\mathcal{I}'$. Thus $V(r, p_1)/V(r, p_0) = F(\gamma^{-1}(r)) = 1 - 1/\gamma^{-1}(r)$ is strictly increasing on $\mathcal{I}'$, as desired.

As we have now established that $\mathcal{F}$ satisfies Conditions 1-3, Proposition 2 gives us a weight function $\lambda : \mathcal{R} \to \mathbb{R}_{>0}$ such that for all $y \in \mathcal{Y}$, the map $r \mapsto \lambda(r) V(r, y)$ is strictly increasing on $\mathring{\mathcal{R}}$. Thus, fixing $r_0 \in \mathring{\mathcal{R}}$, the loss $L(r, y) = \int_{r_0}^r \lambda(r') V(r', y) dr'$ is convex in $r$ for each $y \in \mathcal{Y}$, as noted by Rockafellar [19, Theorem 24.2]. Moreover, as $\lambda > 0$, $L$ elicits $\Gamma$ by Lambert [12, Theorem 6]. $\qquad \square$

While we defer discussion of future work to § 6, it is worth noting here that the argument establishing Condition 3 immediately extends to infinite outcome spaces. Beginning with $p_0, p_1$ being arbitrary distributions, $\Gamma(p_0) \neq \Gamma(p_1)$, one simply observes that $V(\gamma(\alpha), p_0)/V(\gamma(\alpha), p_1) = 1 - 1/\alpha$ by the same logic. The central challenge to such an extension therefore lies in the proof of Proposition 2.

Loosely speaking, when combining Theorem 2 with the existing literature, we conclude that every "nice" elicitable property is additionally convex elicitable. We formalize this in two corollaries, one stated as an implication, and the other given in the style of Steinwart et al. [22, Cor. 9].

**Corollary 1.** *Let $\Gamma : \mathcal{P} \to \mathcal{R}$ be continuous, nowhere-locally-constant, and elicited by a loss $L$ with bounded and continuous first and second derivatives. Suppose also, for all $p \in \mathcal{P}$, that $\frac{d}{dr} \mathbb{E}_{Y \sim p} L(r, Y) = 0$ for at most one $r \in \mathring{\mathcal{R}}$. Then $\Gamma$ is convex elicitable.*

*Proof.* As $L(r, y)$ elicits $\Gamma$ and is differentiable, for all $r \in \mathring{\mathcal{R}}$, and all $p$ with $\Gamma(p) = r$, we must have $\frac{d}{dr} \mathbb{E}_{Y \sim p} L(r, Y) = 0$. By our assumption on the critical points of $L$, we see that $V(r, \delta_y) = \frac{d}{dr} L(r, \delta_y)$ is a bounded and oriented identification function for $\Gamma$, and is continuously differentiable with bounded derivative. Thus, $\mathcal{F} = \{V(r, \delta_y)\}_{y \in \mathcal{Y}}$ satisfies Condition 1, and the result follows from Theorem 2. $\qquad \square$

**Corollary 2.** *Let $\mathcal{P} = \Delta(\mathcal{Y})$ be the probability simplex over $n$ outcomes, and let $\Gamma : \mathcal{P} \to \mathcal{R}$ be a nowhere-locally-constant property with a bounded and nowhere vanishing first derivative, a bounded second derivative, and a differentiable right inverse.[1] Then the following are equivalent:*

1. *For all $r \in \mathcal{R}$, the level set $\{p : \Gamma(p) = r\}$ is convex.*
2. *$\Gamma$ is quasi-monotonic.*
3. *$\Gamma$ is identifiable and has a bounded and oriented identification function.*
4. *$\Gamma$ is elicitable.*
5. *There exists a non-negative, measurable, locally Lipschitz continuous loss function eliciting $\Gamma$.*
6. *$\Gamma$ is convex elicitable.*

*Proof.* We essentially reduce to a similar result of Steinwart et al. [22, Corollary 9]. First, note that the definition of nowhere-locally-constant from Lambert et al. [14] coincides with the definition

of Steinwart et al. [22, Definition 4] in finite dimensions. Second, as our assumptions are stronger than theirs, the equivalence of the first five conditions follows. As convex elicitability implies convex level sets (by the same argument of Lambert [12, Theorem 5], which follows even if $L$ can be infinite on the boundary of $\mathcal{R}$), it then suffices to show that identifiability implies convex elicitability.

By standard arguments, the convexity of the level sets $\{p : \Gamma(p) = r\}$ for $r \in \mathring{\mathcal{R}}$ imply that each level set must be a hyperplane intersected with $\mathcal{P}$. (See e.g. Theorem 1 of [14].) Letting $\hat{p}$ be the right inverse of $\Gamma$, so that $\Gamma(\hat{p}(r)) = r$ for all $r \in \mathcal{R}$, we may define

$$V(r, y) = \nabla_{\hat{p}(r)}\Gamma \cdot (\delta_y - \hat{p}(r)) \,, \tag{3}$$

a form taken from Frongillo and Kash [8, Proposition 18].

Now for any $p$ with $\Gamma(p) = r$, as the level set is a hyperplane intersected with $\mathcal{P}$, we must have $\Gamma(\alpha p + (1-\alpha)\hat{p}(r)) = r$, and we conclude $\nabla_{\hat{p}(r)}\Gamma \cdot (p - \hat{p}(r)) = 0$. (Simply take the derivative with respect to $\alpha$.) Thus, as $\nabla\Gamma \neq 0$, the vector $\nabla_{\hat{p}(r)}\Gamma - \nabla_{\hat{p}(r)}\Gamma \cdot \hat{p}(r)\mathbb{1}$ defines the same hyperplane as $\{p : \Gamma(p) = r\}$, and thus $V$ identifies $\Gamma$. (Here $\mathbb{1} \in \mathbb{R}^{|\mathcal{Y}|}$ denotes the all-ones vector.) That $V$ is also bounded and oriented follows easily from our assumptions. As $V$ has a bounded derivative everywhere by assumption, it satisfies Condition 1, and convex elicitability then follows from Theorem 2. $\qquad\square$

## 4  Proof Sketch and Intuition

We now give a sketch of the construction of the weight function $\lambda$ in Proposition 2. See the Appendix for the full proof. For the purposes of this section, let us simplify our three conditions as follows:

**Condition 1'.**  Every $f \in \mathcal{F}$ is continuously differentiable.

**Condition 2'.**  Each $f \in \mathcal{F}$ has a single zero, and moves from negative to positive.

**Condition 3'.**  When $f > 0 > g$, the ratio $g/f$ is increasing.

*Two function case.*  To begin, let us consider two functions satisfying Conditions 1', 2', and 3', such that $f > 0 > g$ on the interval $\mathcal{I}$. We wish to find some $\lambda : \mathcal{I} \to \mathbb{R}_{>0}$ making both $\lambda f$ and $\lambda g$ strictly increasing. By Condition 3', we know $g/f$ is increasing on $\mathring{\mathcal{I}}$. Let us choose $\lambda$ as follows,

$$\lambda(r) := (-f(r)g(r))^{-1/2} \,. \tag{4}$$

As $-(fg)(r) > 0$ for all $r \in \mathcal{I}$, we have $\lambda(r) > 0$ as well. Moreover, one easily checks that $\lambda f = \sqrt{-f/g}$ and $\lambda g = \sqrt{-g/f}$, which are both increasing as monotonic transformations of $g/f$.

*General case.*  More generally, we wish to find a $\lambda$ such that for all $x \in \mathring{\mathcal{R}}$, $\frac{d}{dx}(\lambda f)(x) > 0$. When $f > 0$, this constraint is equivalent to $\frac{d}{dx}\log(\lambda f)(x) > 0$, which is in turn equivalent to $-\frac{d}{dx}\log\lambda(x) < \frac{d}{dx}\log f(x)$. Similarly, if $f(x) < 0$, then we need $-\frac{d}{dx}\log\lambda(x) > \frac{d}{dx}\log(-f(x))$. Finally, the case $f(x) = 0$ follows easily from Condition 2', as $\frac{d}{dx}f(x) > 0$ and $\lambda > 0$. Combining these constraints, we see that for all $f > 0$ and all $g < 0$, we must have

$$\tfrac{d}{dx}\log(-g(x)) < -\tfrac{d}{dx}\log\lambda(x) < \tfrac{d}{dx}\log f(x) \,. \tag{5}$$

In order for these constraints to be feasible, we must have $\frac{d}{dx}\log(-g(x)) < \frac{d}{dx}\log f(x)$ for all $f < 0 < g$, which is seen to be equivalent to Condition 3' after some manipulation.

Perhaps the most natural way to satisfy constraint (5) is to simply take the midpoint between the maximum lower bound $\underline{m} : \mathcal{R} \to \mathbb{R}$ and minimum upper bound $\overline{m} : \mathcal{R} \to \mathbb{R}$ defined as follows:

$$\overline{m}(x) := -\sup_{g \in \mathcal{F}: g(x)<0} \tfrac{d}{dx}\log(-g(x)) \qquad \underline{m}(x) := -\inf_{f \in \mathcal{F}: f(x)>0} \tfrac{d}{dx}\log(f(x))$$

This yields the following construction (where $r_0 \in \mathring{\mathcal{R}}$ is arbitrary),

$$h(x) = \frac{1}{2}\left(\overline{m}(x) + \underline{m}(x)\right) \,, \quad \lambda(x) = \exp\left(\int_{r_0}^{x} h(z)dz\right) \,, \tag{6}$$

where one notes $h(x) = \frac{d}{dx}\log\lambda(x)$. Provided our three conditions hold, we now have a positive weight function $\lambda$ satisfying the constraint (5), and we conclude that $\lambda f$ is increasing for all $f \in \mathcal{F}$.

Let us observe that our general construction in eq. (6) really is a generalization of the two-function case in eq. (4). That is, we are primarily concerned with the "most decreasing" and "least increasing" functions, which allows us to focus on two functions instead of the entire set $\mathcal{F}$. When we only have two functions $f > 0 > g$, eq. (6) reduces to $h(x) = -\frac{1}{2}\left(\frac{d}{dx}\log(-g(x)) + \frac{d}{dx}\log f(x)\right)$, whence $\lambda(x) = \exp\left(\frac{1}{2}\log(-g(x)f(x))\right) = 1/\sqrt{-g(x)f(x)}$.

**Hurdles and technicalities.** As stated, the above construction has two issues, which we now briefly identify and describe how our proof circumvents. First, in general our functions $f$ will pass through $0$, possibly making $h$ and therefore $\lambda$ unbounded. Recall that we only needed to satisfy eq. (5), and thus rather than taking the midpoint of the lower and upper bounds as in eq. (6), which will diverge whenever one of the bounds diverges, we can always choose $h$ in a slightly more clever manner to be closer to the smallest magnitude bound. See the Appendix for one such construction.

The second problem is that our actual Condition 1 allows for nondifferentiability, which arises in settings of particular interest, like Proposition 1. Fortunately, in the finite-outcome setting, it is essentially without loss of generality to consider continuous $f \in \mathcal{F}$ (see Theorem 1). We can therefore address the finite nondifferentiabilities using continuity arguments, allowing us to focus on the set $\mathcal{I}_c \subseteq \mathcal{I}$ where every $f \in \mathcal{F}$ is continuously differentiable.

# 5 Examples

To illustrate the constructive nature of Theorem 2, we now give two examples. The first is the Beta family scoring rule found in Buja et al. [6, §11] and Gneiting and Raftery [11, §3], which we use to illustrate the construction itself. The second is a simple elicitable property for which the obvious identification function does not give a convex loss; we show how to convexify it.

**1. Beta families.** Consider the Beta family of loss functions discussed in Buja et al. [6], which elicit the mean over outcomes $\mathcal{Y} = \{0, 1\}$, with $\mathcal{R} = [0, 1]$. After some manipulation, one can write the loss and identification function as follows, for any $\alpha, \beta > -1$,

$$L(r, y) = \int_0^r z^{\alpha-1}(1-z)^{\beta-1}(z-y)dz \qquad V(r, y) = r^{\alpha-1}(1-r)^{\beta-1}(r-y) \,.$$

While some choices of the parameters yield convex losses, such as $\alpha = \beta = 0$ (log loss) and $\alpha = \beta = 1$ (squared loss), not all do, e.g. $\alpha = 1/5, \beta = -1/2$.

Applying the two-function construction from Section 4, we choose $\lambda(r) = r^{1/2-\alpha}(1-r)^{1/2-\beta}$, giving the identification function $V'(r, y) = r^{1/2}(1-r)^{1/2}(r-y)$, which is itself in the Beta family with $\alpha = \beta = 1/2$. Intergrating $V'$ yields the following convex loss,

$$L'(r, y) = \int_0^r z^{1/2}(1-z)^{1/2}(z-y)dz = \arcsin(\sqrt{|y-r|}) - \sqrt{r(1-r)} \,, \qquad (7)$$

also discovered by Buja et al., which serves as a intermediary between log and squared loss.

**2. A quadratic property.** Let $\mathcal{Y} = \{1, 2, 3\}$, and $\Gamma(p) = \frac{1-\sqrt{1-4p_1 p_2 + 2p_2^2}}{2p_2}$, where $\Gamma(p) = p_1$ when $p_2 = 0$ for continuity (from L'Hôpital's rule). Here, $p_y$ denotes the probability outcome $y$ is observed. Some of the level sets of $\Gamma$ can be seen in Figure 2. A very natural choice of identification function for $\Gamma$ is $V(r, 1) = r - 1$, $V(r, 2) = \frac{1}{2} + r - r^2$, $V(r, 3) = r$, as one readily verifies. Yet we see in Figure 1(b) that $V(\cdot, 2)$ is not strictly increasing, so the loss given by integrating $V$ will not be convex.

The set $\mathcal{F} = \{V(\cdot, y)\}_{y \in \mathcal{Y}}$ satisfies Conditions 1–3, however, and thus we may use our construction to obtain a positive function $\lambda$ for which $L(r, y) = \int_{r_0}^r \lambda(x)V(x, y)dx$ elicits $\Gamma$ and is convex in $r$. Unfortunately, for this particular example, the construction given in the proof of Proposition 2 produces a somewhat unwieldy function $\lambda$. Fortunately, while our constructed $\lambda$ is guaranteed to make $\lambda f$ monotone for every function $f$ in $\mathcal{F}$, it is generally not unique, and in many cases a simpler choice of $\lambda$ can be found. In particular, our proof shows that *any* function $h$ satisfying the criteria laid out in Claim 1 of the Appendix will lead to suitable choice of $\lambda$; among these criteria are that $h(r) = -\frac{d}{dr}\log\lambda(r)$ must lie between $\underline{m}(r)$ and $\overline{m}(r)$ for all $r$. For practical purposes,

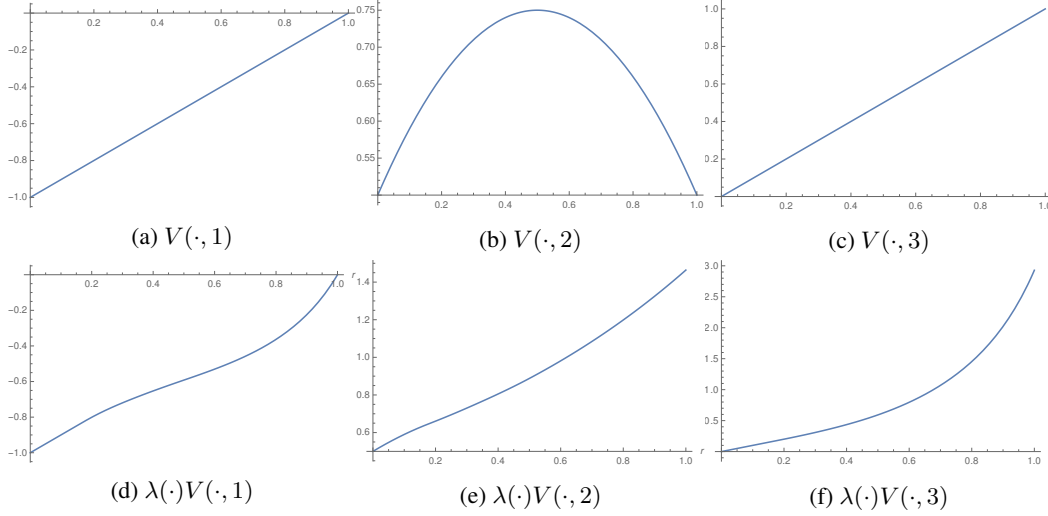

(a) $V(\cdot, 1)$  (b) $V(\cdot, 2)$  (c) $V(\cdot, 3)$

(d) $\lambda(\cdot)V(\cdot, 1)$  (e) $\lambda(\cdot)V(\cdot, 2)$  (f) $\lambda(\cdot)V(\cdot, 3)$

Figure 1: The functions $V(\cdot, y)$ are not always increasing for all $y \in \mathcal{Y}$, but our function $\lambda$ "monotonizes" them, as shown in (d)–(f).

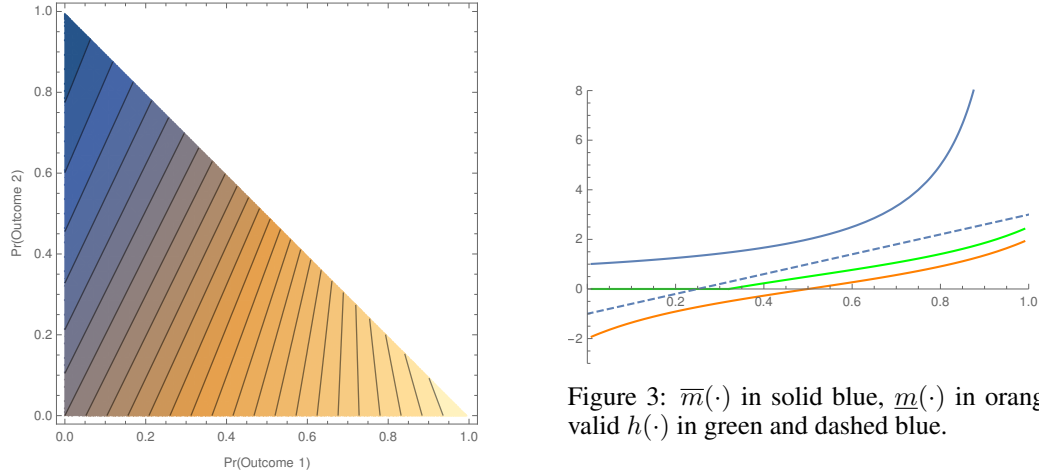

Figure 2: Level sets of $\Gamma$

Figure 3: $\overline{m}(\cdot)$ in solid blue, $\underline{m}(\cdot)$ in orange, valid $h(\cdot)$ in green and dashed blue.

therefore, we may use the following general technique in lieu of the construction given in the proof of Proposition 2:

1. Compute the bounds $\overline{m}(r)$ and $\underline{m}(r)$.

2. Search over some class of practical (e.g. linear) functions for an $h$ which satisfies the criteria of Claim 1.

We illustrate this more practical construction in Figure 3; for the case of our quadratic property, the choice $h(x) = 4x - 1$ (shown as dashed blue) suffices, giving us the simpler $\lambda(r) = \exp(2x^2 - x)$. This choice of $\lambda$ gives

$$\lambda(r)V(r,1) = \exp(2r^2 - r)(r-1)$$
$$\lambda(r)V(r,2) = \exp(2r^2 - r)((1/2) + r - r^2)$$
$$\lambda(r)V(r,3) = r\exp(2r^2 - r),$$

which we can integrate to obtain a convex loss.

# 6 Conclusion and future work

We have shown that all real-valued properties over finite outcomes, which are identified by a mostly-smooth continuous identification function, are convex elicitable. Beyond natural relevance to machine learning, and statistical estimation more broadly, these results bring insights into the area of information elicitation. For example, a generalization of a common prediction market framework, the Scoring Rule Market, is well-defined for any loss function [10, 14]. Yet it is not clear whether practical markets exist for any elicitable property. Among the practical considerations are axioms such as Tractable Trading (TT), which states that participants can compute their optimal trade/action under a budget [2], and Bounded Trader Budget (BTB), which states that traders with arbitrarily small budgets can still fruitfully participate in the market [10]. Our results imply that essentially every continuous real-valued elicitable property over finite outcomes has a market mechanism which satisfies these axioms. There are likely also implications for wagering mechanisms [13] and forecasting competitions [25], among other settings in information elicitation.

There are several avenues for future work, which we outline below.

**Relaxing our conditions.** We believe one could allow $V$ to be smooth almost everywhere. One may still be able to use the fact that $g/f$ is strictly increasing to have an almost-everywhere defined derivative, but again, there are several challenges to this approach.

**Infinite outcomes.** A challenging but important extension would be to allow infinite $\mathcal{Y}$, for example, $\mathcal{Y} = [0, 1] \subseteq \mathbb{R}$. As discussed following Theorem 2, many pieces of our argument extend immediately, such as the argument establishing Condition 3. We believe the key hurdle to such an extension will be in Proposition 2, as several quantities become harder to control. As one example, the definition of $h$ in eq. (6) involves a maximum and minimum which may not be obtained. Extending to infinite outcomes requires the relaxation of our continuity assumption, as many properties of interest have discontinuous identification functions in the infinite-outcome space, like the median.

**Strongly convex losses.** Just as convex loss functions are useful so that empirical risk minimization is a tractable problem, strongly convex losses are even more tractable. Roughly speaking (ignoring the $\log$ transformation), if the gap in eq. (5) is bounded away from zero, $\lambda f$ will be increasing at least as fast as some linear function for all $f$, meaning its integral will be strongly convex. It is not clear what meaningful conditions on $\Gamma$ suffice for this to hold, however, and a full characterization is far from clear. Similarly, characterizations for exp-concave losses would also be interesting.

**Vector-valued properties.** Finally, we would like to extend our construction to vector-valued properties $\Gamma : \mathcal{P} \to \mathbb{R}^k$. In light of our results, this question is only interesting for properties which are not vectors of elicitable properties: if the $k$ components of $\Gamma$ are themselves elicitable, we may construct a convex loss for each, and the sum will be a convex loss eliciting $\Gamma$. Unfortunately, we lack a characterization of elicitable vector-valued properties, so the question of whether all elicitable vector-valued properties are convex elicitable seems even further from reach.

**Acknowledgements.** We would like to thank Bo Waggoner and Arpit Agarwal for their insights and the discussion which led to this project, and we thank Krisztina Dearborn for consultation on analysis results. Additionally, we would like to thank our reviewers for their feedback and suggestions. This project was funded by National Science Foundation Grant CCF-1657598.

## Footnotes

[1] We may identify $\mathcal{P}$ with $\{v \in \mathbb{R}_+^{|\mathcal{Y}|-1} : \sum_{i=1}^{|\mathcal{Y}|-1} v_i \leq 1\}$ so that the derivatives are well defined. In the proof, for ease of notation, we will still write dot products in $\mathbb{R}^{|\mathcal{Y}|}$.

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
