[Reviews · NeurIPS 2018]

Reviewer 1



Summary: This paper attempts to answer an interesting theoretical question regarding property (of a distribution) elicitation. It is already known that 'identifiable' (a set of linear constraints can express distributions with same property value) properties are 'elicitable' (property can be expressed as a minimizer of a loss function in expectation). The central question of this work is under what additional conditions identifiable properties are convex elicitable (convex loss functions elicit those properties). In finite-outcome setting, they show that, under mild smoothness conditions, all real-valued identifiable properties are convex elicitable. The proof relies on a construction of weight (and associated convex loss) function for an identifiable property. Comments: The theoretical result (and the associated loss construction scheme) brings a computational advantage in solving the ERM problem associated with the property. Overall, the paper is well-written with a clear introduction, motivation, and problem statement. The article seems to be technically correct as well.

Reviewer 2



This paper studies which properties of a distribution can be elicited in an incentive compatible way, and previous characterization shows a characterization of such properties. This paper shows that every such property can in fact be elicited by a convex loss functions. This is a very nice result. It also gives new loss functions for some of the properties.

Reviewer 3



Summary: This paper studies elicitation which is defined as the property or statistic of a distribution which can be described as the minimizer of some loss function in expectation. More specifically, the focus of this paper is in studying which properties can be explained as the minimizer of convex loss functions. The authors prove that in the finite outcome setting, under some assumptions, every elicitable real-valued property can also be elicited by a convex loss function. The authors start the paper by providing an overview of the prior literature and go over their assumptions. Then, they state the theorem and give some intuition on how the proof goes through. Finally, they conclude the paper by providing two examples and applying their constructive technique to yield corresponding convex losses. Evaluation: The paper is nicely written and well-motivated. As authors also mention, in many practical applications, dealing with convex losses is easier than general loss functions as various techniques can be applied for finding their minimum. Authors have addressed the limitation of their method in their discussion, however I really think that the generalization to infinite outcome space is very important. In other words, I believe that this limitation is very restrictive and needs to be better addressed. For this more general case, authors could have either provided some proofs (positive results) with maybe some additional assumptions, or they could maybe provide some counter-examples (negative results). Furthermore, even in the case of finite outcome setting, the second example in section 5 shows that even the constructive way of their theorem (or more precisely, the constructed weight function \lambda) produces a very complex \lambda. So other than a more simpler setting of Example 1, the proof is not practically constructive. I understand that the authors were able to address this concern for Example 2 (which I think it is nice), but still this construction might be far from practical purposes in general. Based on these points, I think the paper is in a very good shape in terms of clarity, but can be further improved in technical and also practical aspects. AFTER REBUTTAL: The authors have addressed all my concerns and I am willing to increase my score from 6 to 8. I believe the idea of convex elicitation is very important and this paper makes a huge contribution to the field. Hopefully, the result can be extended to the infinite outcome space in the near future.